# Differential Adjuvant Activity by Flagellins from *Escherichia coli*, *Salmonella enterica* Serotype Typhimurium, and *Pseudomonas aeruginosa*

**DOI:** 10.3390/vaccines12111212

**Published:** 2024-10-25

**Authors:** Shengmei Pang, Mei Liu, Longlong Wang, Mingqing Shao, Guoqiang Zhu, Qiangde Duan

**Affiliations:** 1College of Veterinary Medicine, Yangzhou University, Yangzhou 225009, China; dz120210012@stu.yzu.edu.cn (S.P.); mz120221581@stu.yzu.edu.cn (M.L.); w13028032885@163.com (L.W.); mz120231712@stu.yzu.edu.cn (M.S.); yzgqzhu@yzu.edu.cn (G.Z.); 2Jiangsu Co-Innovation Center for Prevention and Control of Important Animal Infectious Diseases and Zoonoses, Yangzhou 225009, China; 3Joint Laboratory of International Cooperation on Prevention and Control Technology of Important Animal Diseases and Zoonoses of Jiangsu Higher Education Institutions, Yangzhou University, Yangzhou 225009, China

**Keywords:** flagellin, adjuvant activity, TLR5, innate immune

## Abstract

(1) **Background**: The adjuvant properties of flagellin from various bacterial species have been extensively studied; however, a systematic comparison of the immunoadjuvant effects of flagellins from different bacterial species is lacking. This study aims to analyze the amino acid sequences and structural features of flagellins from *Escherichia coli* (FliC*_E.C_*), *Salmonella enterica* serotype Typhimurium (FliC*_S.T_*), and *Pseudomonas aeruginosa* (FliC*_P.A_*), and to evaluate their adjuvant activities in terms of Toll-like receptor 5 (TLR5) activation, antibody production, and cytokine responses in a murine model. (2) **Methods**: Bioinformatics analysis was conducted to compare the amino acid sequences and structural domains (D0, D1, D2, and D3) of flagellins from the three bacterial species. PyMol atomic models were used to confirm structural differences. Toll-like receptor 5 (TLR5) activation assays were performed to measure IL-8 and TNF-α production in vitro. The IgG antibody titers against the model antigen FaeG and cytokine responses, including IL-4 and TNF-α secretion were evaluated in a murine model. (3) **Results**: Bioinformatics analysis revealed that the D0 and D1 domains are highly conserved, whereas the D2 and D3 domains exhibit significant variability across the three species. Structural analysis via PyMol confirmed these differences, particularly in the D2 and D3 domains. TLR5 activation assays showed that FliC*_S.T_* and FliC*_P.A_* induced higher levels of IL-8 and TNF-α production compared to FliC*_E.C_*, indicating species-specific variations in TLR5 activation. In the murine model, FliC*_S.T_* as an adjuvant produced higher antibody titers against FaeG and increased IL-4 secretion in splenocytes compared to FliC*_E.C_* and FliC*_P.A_*. FliC*_P.A_* induced higher TNF-α expression than FliC*_S.T_* and FliC*_E.C_*, suggesting FliC*_S.T_* and FliC*_P.A_* are more effective at inducing T-cell responses. (4) **Conclusions**: This study highlights the potential of FliC*_S.T_* and FliC*_P.A_* as potent vaccine adjuvants. The results provide insights into the structure–function relationships of these flagellins and support their application in enhancing immune responses against diverse pathogens.

## 1. Introduction

The global development of vaccines remains a pivotal strategy in the prevention and control of infectious diseases. However, the efficacy of many vaccine antigens is often compromised due to their inherent low immunogenicity. This challenge underscores the urgent need for effective adjuvants and substances designed to enhance the immune response to vaccine antigens. These adjuvants typically function by activating both the innate and adaptive immune systems, thereby amplifying the overall protective efficacy of vaccines.

In recent years, there has been growing interest in microbial components as adjuvants, with bacterial flagellin emerging as a promising candidate due to its ability to potentiate immune responses [1,2,3]. Flagellin, the principal structural protein of bacterial flagella, is prevalent in many Gram-negative and some Gram-positive bacteria. Structurally, flagellin is composed of four basic domains (D0, D1, D2, and D3), which fold into a shape resembling the Greek letter “Γ”. The conserved N and C terminal chains flank a central hypervariable domain, forming the core of the filament. The D0 and D1 domains are highly conserved among bacterial species, playing crucial roles in Toll-like receptor 5 (TLR5) activation and flagellar assembly [4]. Specifically, the D0 domain is responsible for TLR5 signaling, while the D1 domain is crucial for filament formation and TLR5 binding [5]. The highly variable D2/D3 domains, located on the outer surface of the filament, are associated with immune evasion and antibody interaction in flagellated pathogens [6]. Most antibody-flagellin interactions target the D3 domain, which has documented the H-antigen subtypes immunostimulatory activity and is essential for innate immune recognition [7,8].

Beyond its role in bacterial motility, chemotaxis, invasion, and colonization, flagellin is instrumental in activating host immune responses through interactions with pattern recognition receptors (PRRs) [9,10,11,12,13]. Notably, its interaction with TLR5 has positioned flagellin as a compelling candidate for use as an immunoadjuvant and a potential vaccine component. Significant advancements have been made in studying flagellin as an immune adjuvant [2]. Research indicates that flagellin, when mixed or fused with antigenic proteins, can markedly enhance antigen-specific immune responses, encompassing both humoral and cellular immunity [14,15,16,17,18]. The underlying mechanism is primarily attributed to flagellin’s interaction with TLR5, which activates the MyD88-dependent signaling pathway [19,20,21]. This activation leads to the activation of nuclear factor kappa B (NF-κB) and the release of pro-inflammatory cytokines, including IL-6, IL-10, and TNF-α. These cytokines are crucial in stimulating both innate and adaptive immune responses [21,22]. In addition to TLR5 interaction, flagellin is recognized by intracellular receptors such as NOD-like receptor protein 4 (NLRC4) and inflammasome receptors NAIP5/6, further triggering pro-inflammatory signaling pathways to clear infections effectively [23].

The broad development involves the use of flagellin fusion proteins or their combination with antigens to exert adjuvant effects. This approach has been widely utilized in developing vaccines for influenza, tuberculosis, and cancer, among others [24,25,26]. Furthermore, flagellin-based vaccines have shown promise in generating mucosal immunity by stimulating IgA production in mucosal tissues via mucosal routes. However, clinically approved vaccines that utilize flagellin as an adjuvant have not yet been authorized. These adjuvants are currently being tested in preclinical and clinical trials, such as VAX125 (STF2-HA in an influenza vaccine candidate) and STF2.4xM2e (M2e fused to the C-terminus of the full-length sequence of *Salmonella enterica* serotype Typhimurium flagellin phase 2 (fljB) in influenza vaccines) [27,28]. Despite flagellin’s considerable potential as an adjuvant, comprehensive comparative studies on the adjuvant activities of flagellins from different bacterial species remain limited [29]. While previous studies have confirmed that flagellin can significantly enhance immune responses to vaccine antigens, the extent to which flagellins from various sources differ in their adjuvant activities is still not fully understood. The immunostimulatory properties of flagellin are intricately linked to its amino acid sequence and three-dimensional structure, which may vary based on the bacterial source. Furthermore, differences in TLR5 binding sites potentially influence these properties. Given research on variations in TLR5 activity, antibody titers, and pro-inflammatory cytokine expression induced by flagellins from different bacterial sources is limited, further investigation is necessary.

This study seeks to systematically analyze and evaluate the adjuvant activities of flagellins derived from different bacterial species, including *Escherichia coli* (*E. coli*), *S.* Typhimurium, and *Pseudomonas aeruginosa* (*P. aeruginosa*). By comparing these flagellins’ effects on TLR5 activation, antibody titers, and cytokine expression, this research aims to elucidate the potential and limitations of flagellin as an immunoadjuvant, providing a scientific foundation for developing new vaccines. Furthermore, the study will explore the application potential of flagellin across various immune response models, offering valuable insights for future vaccine design targeting diverse pathogens.

## 2. Materials and Methods

### 2.1. Bacterial Strains and Plasmids

Genomic DNA from Enterohemorrhagic *E. coli* (EHEC) EDL933, *S*. Typhimurium, and *P*. *aeruginosa* served as templates for amplifying the *fliC_E.C_*, *fliC_S.T_*, and *fliC_P.A_* genes, respectively. *E. coli* strain C83902 was used as the template for PCR amplification of the *faeG* gene. *E. coli* strains DH5α and BL21 (DE3) were obtained from TIANGEN, China, while the pET28α (+) vector was sourced from Novagen, Glendale, CA, USA.

### 2.2. Gene Amplification and Construction of Recombinant Plasmids

The genes encoding *fliC_E.C_*, *fliC_S.T_*, *fliC_P.A_*, and *faeG* were amplified from genomic DNA of EHEC EDL933, *S.* Typhimurium, *P*. *aeruginosa*, and *E. coli* strain C83902 respectively, uing the specific primers listed in Table 1. The purified PCR products were ligated into the digested expression vector pET28α (+) using *BamH*I and *Sal*I/*Sac*I restriction enzymes (NEB, Ipswich, MA, USA). The recombinant plasmids, pET28α-*fliC_E.C_*, pET28α-*fliC_S.T_*, pET28α-*fliC_P.A_*, and pET28α-*faeG*, were confirmed by PCR and DNA sequencing.

### 2.3. Sequence Analysis and Modeling of Flagellins

A comparative analysis of sequence homology was conducted using the Lasergene software MegAlign program version 17.3 (DNAStar, Madison, WI, USA). Multiple sequence alignment of *fliC_E.C_*, *fliC_S.T_*, and *fliC_P.A_* was performed using DNAMAN software version 10.0 with the Clustal W method. The monomer models of the FliC*_E.C_*, FliC*_S.T_*, and FliC*_P.A_* were predicted and analyzed using the AlphaFold 3 server [30]. These models were then simulated and visualized with PyMol version 3.0 (https://pymol.org/2/), revealing structural differences.

### 2.4. Expression and Purification of Recombinant Flagellin Proteins

The recombinant FliC*_E.C_*, FliC*_S.T_*, FliC*_P.A_*, and FaeG proteins were expressed and purified as previously described [31]. The recombinant plasmids were transformed into *E. coli* BL21 (DE3) to express the target proteins with a His-tag. Protein expression was induced with isopropyl-β-D-1-thiogalactoside (IPTG) at a final concentration of 1 mM for 4 h at 37 °C when the OD_600nm_ reached 0.6–0.8. The expression of target proteins was confirmed by sodium dodecyl sulfate–polyacrylamide gel electrophoresis (SDS-PAGE). Following lysis with an ultrasonic cell disruptor, recombinant flagellins in the supernatant were purified using Protino Ni-TED 2000 packed columns (MACHEREY–NAGEL, Düren, Germany) according to the manufacturer’s recommendations. To quantify the purified protein concentration, bovine serum albumin (BSA) was used as a standard with the Molecular Imager Gel Doc^TM^ XR+ system (BIO-RAD, Hercules, CA, USA). The presence of the target proteins was confirmed using anti-FliC*_E.C_* mouse serum (1:8:000), anti-FliC*_S.T_* mouse serum (1:8000), and anti-FliC*_P.A_* mouse serum (1:8000), which were obtained by immunizing mice with crude extracts of flagellins derived from EHEC EDL933, *S*. Typhimurium, and *P. aeruginosa*, with anti-F4 mouse serum (1:10,000), obtained by immunizing mice with crude extracts of F4 fimbriae derived from *E. coli* strain C83902 as primary antibodies, with HRP-conjugated goat anti-mouse IgG (1:10,000) as the secondary antibody (ABclonal, Wuhan, China). Reactive bands were visualized using the chemiluminescence kit ECLUltra (New Cell and Molecular Biotech, Suzhou, China).

### 2.5. Endotoxin Removal and Measurement

The removal of endotoxins from the FliC*_E.C_*, FliC*_S.T_*, FliC*_P.A_*, and FaeG recombinant proteins was accomplished using Pierce™ High-Capacity Endotoxin Removal Resin (Thermo Fisher Scientific, Waltham, MA, USA) [32]. The endotoxin levels of the purified proteins were determined and confirmed to be lower than 0.05 EU/mL using the Pierce™ Chromogenic Endotoxin Quant Kit (Thermo Fisher Scientific).

### 2.6. TLR5 Bioactivity Assay of Flagellins In Vitro

The TLR5-expressing cell line Caco-2 (ATCC HTB-37), which is derived from human colon carcinoma tissue, was utilized to evaluate the TLR5-ligand activities of flagellins, following the established protocol [33]. Cells were stimulated with 5 μg/mL of purified recombinant FliC*_E.C_*, FliC*_S.T_*, and FliC*_P.A_* for 6 h in a 6-well plate, with a seeding density of 5 × 10^5^ cells per well (Corning, Corning, NY, USA). As a negative control, cells were treated with culture medium alone. The concentrations of IL-8 and TNF-α in the supernatants were quantified using commercial human IL-8 and TNF-α ELISA kits (NeoBioscience, Beijing, China). Cytokine concentrations were determined via standard curves, with actual values derived by subtracting the background absorbance from the absorbance values obtained from the standard curve.

### 2.7. Animal Immunization Regimens

The animal experiments were approved by the Yangzhou University Institutional Animal Care and Use Committee (reference number 202103034) and conducted in compliance with the National Institute of Health guidelines for the ethical use of animals in China. Thirty specific pathogen-free (SPF) female BALB/c mice, aged 6–8 weeks, were obtained from the Animal Experiment Center of Yangzhou University. The mice were randomly divided into five groups, each consisting of 6 mice. The FaeG protein, used as a model antigen in this study, is the major subunit of F4 fimbriae, which is essential for the adhesion of porcine enterotoxigenic *Escherichia coli* (ETEC) that produce F4 fimbriae to host intestinal epithelial cells. Mice in groups 1 to 3 were subcutaneously inoculated with 50 μg of recombinant flagellins (FliC*_E.C_*, FliC*_S.T_*, or FliC*_P.A_*) co-mixed with 50 μg of FaeG protein (flagellin/FaeG 1:1, *w*/*w*). Mice in group 4 were subcutaneously inoculated with 50 μg of FaeG protein emulsified in an equal volume of Freund’s adjuvant. The two control groups received either an equivalent volume of sterile PBS or FaeG protein alone, administered by the same route at the same dose. All groups received booster immunizations at 2-week intervals under the same conditions. Immune sera were collected from the infra-orbital plexus before each immunization.

### 2.8. The Serum Anti-FaeG Specific IgG Antibody Response

The enzyme-linked immunosorbent assay (ELISA) was performed to quantify the IgG titers in individual mouse serum samples [34]. ELISA plates (Corning, USA) were coated with purified K88ac fimbriae (500 ng per well), diluted in carbonate buffer (pH9.6), and incubated overnight at 4 °C. Subsequently, the plates were then blocked with 10% skim milk buffer for 1 h at 37 °C. Each mouse serum sample was serially diluted from 1:400 to 1:25,600 with PBST (PBS/0.05% Tween-20) (pH7.4) in 100 µL per well and incubated at 37 °C for 1.5 h, followed by three washes with PBST. Horseradish peroxidase (HRP)-labeled goat anti-mouse IgG (Sigma, St. Louis, MO, USA) was added at a 1:5000 dilution and incubated at 37 °C for 1 h, followed by three washes with PBST. After washing, 3,3′-5,5′-tetramethylbenzidine (TMB) (Beyotime, Nantong, China) substrate was added for a 30 min incubation at room temperature. The absorbance of each well was measured at 650_nm_ using a microplate reader to determine the antibody titers.

### 2.9. Cytokine Expression Measured in Mouse Splenocytes

The spleen tissues from the FliC*_EC_*, FliC*_ST_*, and FliC*_PA_* immunized groups were carefully disaggregated in PBS using a syringe plunger and subsequently suspended in a Petri dish. The resultant splenocytes were passed through a 75 μm cell strainer to generate a single-cell suspension and treated with red blood cell (RBC) Lysis Buffer (Absin, Shanghai, China) to remove redundant RBCs under sterile conditions. The single-cell suspension was then aliquoted into a 6-well plate (2.5 × 10^6^ cells/well) (NEST, Shanghai, China) containing RPMI 1640 medium (Gibco, Waltham, MA, USA) supplemented with 0.1% Fetal Bovine Serum (FBS) (Lonsera, Suzhou, China) and cultured overnight. To re-stimulate the spleen cells, purified FaeG protein was added at a concentration of 50 µg/mL, and the cells were incubated at 37 °C for 72 h in vitro. Unstimulated wells and the culture medium served as negative controls and blanks, respectively. Following incubation, the culture supernatant was separated from the cells. The levels of inflammatory factors were assessed using quantitative real-time PCR (qRT-PCR) and ELISA kits (Neobioscience, China) to quantify the mRNA and protein levels, respectively.

Total mRNA from the stimulated splenocytes was extracted with TRNzol solution (TianGen, Beijing, China) and reverse-transcribed into complementary DNA (cDNA) using the FastKing gDNA Dispelling RT SuperMix Kit (TianGen, Beijing, China). The resulting cDNA was utilized as a template for qRT-PCR. Specific primers for *Il4*, *Tnfα*, and the reference gene *GADPH*, as detailed in Table 1, were employed. The amplification protocol involved an initial denaturation at 95 °C for 5 min, followed by 40 cycles of 95 °C for 10 s and 60 °C for 30 s. Melting curve analysis was performed under the following conditions—95 °C for 15 s, 60 °C for 60 s, and 95 °C—for 15 s to confirm primer specificity. Each sample was analyzed in triplicate, and results were calculated as relative fold changes using the 2^−ΔΔCt^ method.

The concentrations of IL-4 and TNF*-α* in the splenocyte supernatants were measured using commercial mouse ELISA kits (Neobioscience, China) in accordance with the manufacturer’s instructions [35].

### 2.10. Statistical Analysis

The experimental results were analyzed using GraphPad Prism version 9.0 (GraphPad Software, San Diego, CA, USA). Data are presented as the mean ± SEM for each experiment. Statistical differences between groups were assessed using a one-way ANOVA. Statistical significance compared to the control group is indicated by asterisks (*, *p* < 0.05, **, *p* < 0.01).

## 3. Results

### 3.1. Bioinformatics Analysis of Flagellin from Three Different Bacterial Species

The amino acid sequences from the three species were compared and analyzed using ClustalW in DNAMAN software. As depicted in Figure 1A, their extreme amino- and carboxyl-terminals (D0 and D1 domains) exhibited relatively conserved and identical sequences. Variations were mainly localized in the central region (D2 and D3 domains), affecting the length and primary structure of the amino acids. Specifically, the homology and similarity of amino acid sequences among the three species varied from 49.7% to 67.7% (Figure 1B). The TLR5 binding site (89–96th aa) identities ranged from 55.6% to 77.8%, with FliC*_E.C_* showing greater homology (77.8%) with FliC*_S.T_*, and a 55.6% similarity between FliC*_S.T_*_,_ and FliC*_P.A_* (Figure 1C). Additionally, residues Q89, R90, L94, and E114, which are important for TLR5 activation, are completely identical among the three species [36].

The atomic models rendered with PyMol illustrate that the scaffolds of the three bacterial flagellin monomer structural architectures have distinct configurations (Figure 1D–F). The shapes of the protein monomers FliC*_S.T_* and FliC*_E.C_*resemble the capital letter “Γ”. All the D0 and D1 domains share high structural homology characterized by α-helix coiled coils, whereas the D2 and D3 domains are heterogeneous among the three flagellins, displaying entirely different folds and turns, consistent with the high sequence variations. Specifically, the D2 and D3 domains comprise higher β-sheets with handles and random coils. Moreover, Q89, R90, L94, and E114 residues are localized on the surface of the three backbones and are critical for TLR5 interaction.

### 3.2. Expression, Purification, and Identification of Flagellins

The PCR products of *fliC_E.C_*, *fliC_S.T_*, and *fliC_P.A_* were successfully amplified and cloned into the pET28α (+) expression vector to construct recombinant plasmids, which were subsequently transformed into *Escherichia coli* BL21 (DE3) for protein expression. The three recombinant flagellins were highly expressed in a soluble form within the *E. coli* cells. After purification using Protino Ni-TED 2000 packed columns (MACHEREY–NAGEL, Germany), sodium dodecyl sulfate-–polyacrylamide gel electrophoresis (SDS-PAGE) analysis revealed single, specific bands corresponding to FliC*_E.C_*, FliC*_S.T_*, and FliC*_P.A_*, with molecular weights of approximately 60 kDa, 52 kDa, and 40 kDa, respectively, aligning with the calculated sizes of these proteins (Figure 2A). Furthermore, Western blot analysis demonstrated that the recombinant FliC*_E.C_*, FliC*_S.T_*, and FliC*_P.A_* proteins were recognized by anti-FliC*_E.C_*, anti-FliC*_S.T_*, and anti-FliC*_P.A_* mouse polyclonal antibodies, respectively. This confirmed the good reactogenicities of these purified recombinant flagellins (Figure 2B–D).

### 3.3. Differential TLR5 Activation by Flagellins from Various Species

To evaluate the TLR5 activation abilities of the purified recombinant FliC*_E.C_*, FliC*_S.T_*, and FliC*_P.A_*, the concentrations of IL-8 and TNF-α were measured in the supernatant of Caco-2 cells following stimulation by these three flagellins. ELISA results showed that the IL-8 and TNF-α levels were significantly higher in the flagellin-treated groups compared to the untreated group (*p* < 0.01). The concentration of IL-8 was significantly higher in the FliC*_P.A_* stimulation group compared to the FliC*_E.C_* and FliC*_S.T_* stimulation groups, whereas no difference was observed between the FliC*_E.C_* and FliC*_S.T_* stimulation groups (Figure 3A). Similarly, the TNF-α concentration was significantly higher in the FliC*_P.A_* and FliC*_S.T_* stimulation groups compared to the FliC*_E.C_* stimulation group, with no difference between the FliC*_S.T_* and FliC*_P.A_* stimulation groups (Figure 3B). The findings suggest that the flagellins from different species may exhibit varying TLR5 activation capabilities in vitro.

### 3.4. Enhanced Antibody Response Using FliC_E.C_, FliC_S.T_, and FliC_P.A_ as Adjuvants

To assess the adjuvant efficacy of FliC*_E.C_*, FliC*_S.T_*, and FliC*_P.A_*, mice were immunized with a mixture of the model antigen FaeG (a major subunit of F4 fimbriae) and each of the flagellins as adjuvants. Freund’s Adjuvant, standard adjuvants widely used in immunological studies and vaccine development, served as positive controls. The immunization schedule is illustrated in Figure 4A. Anti-FaeG antibody titers were measured by ELISA. Results showed that all immunization groups, except the PBS group, produced anti-FaeG antibodies one week after the first immunization. Significant IgG responses were observed, with progressively elevated specific anti-FaeG IgG levels after each immunization (Figure 4B). The response peaked in the sixth week and remained stable over time. The groups immunized with FliC*_E.C_*, FliC*_S.T_*, and FliC*_P.A_* as adjuvants exhibited higher levels of anti-IgG antibody compared to the FaeG alone and PBS groups (*p* < 0.001). The IgG titers in the FaeG + FliC*_E.C_* (3.65 ± 0.11) and FaeG + FliC*_P.A_* (3.66 ± 0.11) groups were comparable, though both were lower than the positive control group FaeG + CFA (3.81 ± 0.20) (*p* > 0.05). The FliC*_S.T_* adjuvant group (3.81 ± 0.11) demonstrated a higher anti-IgG antibody titer than the FliC*_E.C_* (3.65 ± 0.11) and FliC*_P.A_* (3.66 ± 0.11) groups, but the difference was not statistically significant (*p* > 0.05) (Figure 4C). While the adjuvant effects of FliC*_E.C_* and FliC*_P.A_* were slightly weaker than Freund’s adjuvant, they still demonstrated robust adjuvant activity. Notably, FliC*_S.T_* showed adjuvant efficacy comparable to Freund’s Adjuvant, indicating its strong potential for immune enhancement applications.

### 3.5. Enhanced IL-4 and TNF-α Responses with FliC_E.C_, FliC_S.T_, and FliC_P.A_ Adjuvants

The levels of IL-4 and TNF-α were measured in splenocytes and their cultured supernatants from immunized mice stimulated with the model antigen FaeG, using qRT-PCR and ELISA for mRNA and protein levels, respectively. Co-administration of FaeG with the flagellins resulted in higher mRNA levels of the cytokines IL-4 and TNF-α compared to the FaeG alone group, with varying performance relative to the positive control group FaeG + CFA. IL-4 expression was significantly higher in the FaeG + FliC*_S.T_* group (4.36 ± 0.43-fold) compared to the FaeG + FliC*_E.C_* (1.94 ± 0.2-fold), FaeG + FliC*_P.A_* (2.04 ± 0.17-fold), FaeG antigen alone (1.53 ± 0.14-fold), and FaeG + CFA (2.83 ± 0.06-fold) groups (*p <* 0.01) (Figure 5A). Conversely, TNF-α induction was markedly higher in the FaeG + FliC*_P.A_* group (24.14 ± 0.52-fold) compared to the FaeG + FliC*_E.C_* (4.28 ± 0.15-fold), FaeG + FliC*_S.T_* (6.51 ± 0.53-fold), FaeG antigen alone (2.0 ± 0.18-fold), and FaeG + CFA (7.55 ± 0.93-fold) groups (*p <* 0.001) (Figure 5B). No significant difference in IL-4 mRNA levels was observed between the FaeG + FliC*_E.C_* and FaeG + FliC*_P.A_* groups; however, the TNF-α level was higher in the FaeG + FliC*_S.T_* group compared to the FaeG + FliC*_E.C_* group. ELISA results for IL-4 and TNF-α levels were largely consistent with the mRNA findings. At the translational level, IL-4 secretion was significantly higher in the FaeG + FliC*_S.T_* group compared to the FaeG + FliC*_E.C_*, FaeG + FliC*_P.A_*, FaeG antigen alone, and FaeG + CFA groups (*p* < 0.01) (Figure 5C). TNF-α production was significantly higher in the FaeG + FliC*_P.A_* group compared to the FaeG + FliC*_E.C_*, FaeG + FliC*_S.T_*, FaeG antigen alone, and FaeG + CFA groups (*p* < 0.01) (Figure 5D). Co-administration of FaeG with flagellins significantly enhanced the mRNA levels of IL-4 and TNF-α, demonstrating varying immune responses compared to the FaeG alone and positive control groups. Notably, the FaeG + FliC*_S.T_* group showed the highest IL-4 expression, while TNF-α levels were markedly elevated in the FaeG + FliC*_P.A_* group. Compared to Freund’s adjuvant (FaeG + CFA), FliC*_S.T_* exhibited a superior induction of IL-4, while FliC*_P.A_* showed significantly higher TNF-α production, suggesting that both flagellins possess potent adjuvant activities in eliciting and modulating T-cell responses, offering potential advantages over traditional adjuvants. These findings highlight the potential of these flagellins as effective alternatives to traditional adjuvants for enhancing immune responses in vaccine development.

## 4. Discussion

This study presents a detailed comparative analysis of the adjuvant properties of flagellins derived from the enterobacteriaceae such as *E. coli* and *S.* Typhimurium, and non-enteric γ-proteobacteria such as *P. aeruginosa*. Our results underscore the significant potential of these flagellins as powerful immunoadjuvants, particularly emphasizing their ability to activate TLR5 and significantly potentiate immune responses. Importantly, we elucidate distinct differences in the immunostimulatory efficacy among the flagellins of these three bacterial species, which could lead to tailored applications in vaccine development, aligning with the specific immunological requirements of different pathogens.

Bacterial flagellin is composed of four domains, D0 through D3. Among these, the D0 and D1 domains are notably conserved across a wide range of bacterial species, in contrast to the D2 and D3 domains, which exhibit considerable variability [37,38]. In line with previous reports, comparative analysis of the amino acid sequences of flagellins from *E. coli*, *S.* Typhimurium, and *P. aeruginosa* revealed that while the D0 and D1 domains were relatively conserved across the three species, substantial variability was observed in the D2 and D3 domains [9,39,40]. This variability in the D2 and D3 domains is likely responsible for the distinct structural configurations of the flagellins, as demonstrated by the PyMol—rendered atomic models. The conserved D0 and D1 domains are crucial for TLR5 activation, particularly the residues 89–96 located in the D1 domain, which constitute the dominant TLR5 binding and activating site [41,42]. Notably, our results showed that this motif exhibited varying degrees of homology, with FliC*_E.C_* showing the highest similarity to FliC*_S.T_*. This suggests that even minor sequence differences in critical binding regions can significantly influence the overall structural conformation and, consequently, the biological activity of the flagellin proteins. Supporting these structural findings, the functional assays demonstrated that all three flagellins could activate TLR5, as evidenced by the increased production of IL-8 and TNF-α in Caco-2 cells. However, the extent of this activation varied significantly. Both FliC*_S.T_* and FliC*_P.A_* were more potent than FliC*_E.C_* in stimulating IL-8 and TNF-α production. The differential activation of TLR5 by these flagellins underscores the importance of structural conformation in determining the immunostimulatory potential of flagellins from different bacterial species [43,44].

Flagellins from multiple bacterial species have been demonstrated to possess potent adjuvant activity when mixed or fused with antigenic proteins [45,46]. In this study, we evaluated the adjuvant properties of FliC*_E.C_*, FliC*_S.T_*, and FliC*_P.A_* by assessing the antibody response to the model antigen FaeG in immunized mice. FaeG was chosen as the model antigen because it is well-characterized and known to elicit immune responses, allowing for a reliable evaluation of the adjuvant efficacy of the flagellin proteins. Our results indicated that all three flagellins effectively enhanced the production of specific IgG antibodies, with FliC*_S.T_* showing a marginally higher, though not statistically significant increase in IgG titers compared to the other two flagellins. This finding suggests that FliC*_S.T_* may possess slightly superior adjuvant properties, although the differences were not substantial. As an adjuvant, bacterial flagellin not only induces a robust humoral immune response but also induces a strong cellular immune response [47]. The enhanced IL-4 and TNF-α responses further supported the potent adjuvant activity of these flagellins, particularly FliC*_S.T_* and FliC*_P.A_*, which were more effective in eliciting T-cell responses. The findings of this study have important implications for the design of next-generation vaccines. The ability of FliC*_P.A_* to robustly induce IL-8 production, coupled with its strong TNF-α response, makes it a promising candidate for inclusion in vaccines targeting pathogens where a strong inflammatory response is desired. Conversely, the superior IL-4 induction by FliC*_S.T_* suggests its potential utility in vaccines to elicit a balanced Th2 immune response. The structural insights gained from this study also provide a foundation for the rational design of flagellin-based adjuvants, where specific domains could be engineered to enhance desired immunological outcomes.

Our study aligns with existing reports on the adjuvant properties of bacterial flagellins, which have been shown to enhance vaccine-induced immune responses primarily through TLR5 activation [3,48]. However, our research extends beyond previous studies by offering a more detailed comparative analysis of flagellins derived from different bacterial species. This analysis provides new insights into the structural and functional variations that underlie their diverse adjuvant activities. A significant contribution of this study is the identification of conserved and variable regions within flagellins that critically influence their interaction with TLR5 and, consequently, their immunostimulatory effects. This finding also advances our understanding of the structure–function relationship in flagellins, a crucial aspect of the rational design of novel adjuvants. Despite these promising findings, further research is warranted to fully elucidate the mechanisms driving the differential adjuvant activities observed among flagellins from various bacterial sources. Future investigations should aim to precisely map the TLR5 binding sites and identify specific amino acid residues that enhance the immunogenicity of certain flagellins. Additionally, exploring the synergistic potential of combining flagellins with other adjuvants or immunomodulators could lead to further improvements in vaccine efficacy [49]. Examining the application of these findings across different animal models and against various pathogens will be essential for advancing these insights toward clinical applications. Moreover, given the structural variability observed in the D2 and D3 domains of flagellins, there is a compelling rationale to explore the engineering of chimeric flagellins that integrate the most immunogenic elements from different bacterial species [16]. Such an approach could pave the way for the development of next-generation adjuvants with optimized immunostimulatory properties, potentially offering enhanced and tailored immune responses for a wide range of vaccines.

## 5. Conclusions

This study offers valuable insights into the differential adjuvant activities of flagellins derived from *E. coli*, *S.* Typhimurium, and *P. aeruginosa*. Our results demonstrate that the three novel and mild adjuvants—FliC*_S.T_*, FliC*_E.C_*, and FliC*_P.A_*—exhibited significant immune enhancement, with FliC*_S.T_* showing efficacy comparable to Freund’s adjuvant. This finding indicates that these novel adjuvants have the potential to compete with traditional adjuvants and serve as promising tools for future animal vaccine development. The findings underscore the substantial potential of these flagellins as vaccine adjuvants, emphasizing the critical importance of selecting the appropriate flagellin based on its unique structural and functional properties. This research not only deepens our understanding of the immunostimulatory mechanisms of flagellins but also establishes a foundation for future investigations aimed at optimizing flagellin-based adjuvants for broader applications in vaccines targeting a wide array of infectious diseases. The implications of this work extend to the rational design of next-generation vaccines, where the tailored use of flagellins could enhance immune responses and improve vaccine efficacy across diverse pathogenic challenges.

## Figures and Tables

**Figure 1 vaccines-12-01212-f001:**
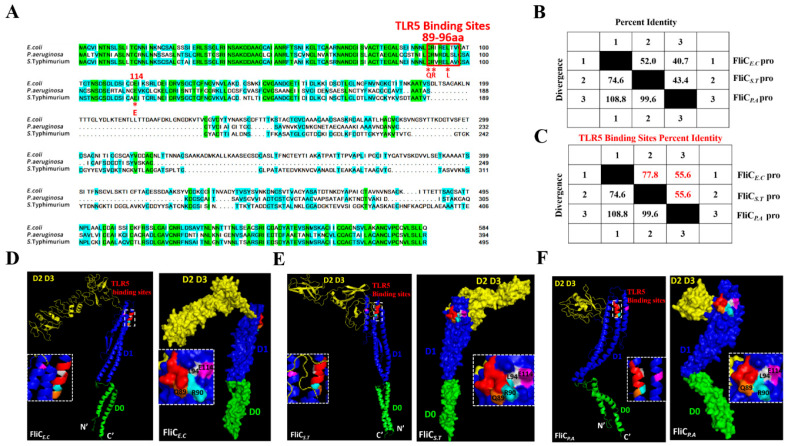
Amino acid sequence comparison and analysis of nucleotide and protein homology of FliC derived from various species. (**A**) An alignment of FliC*_E.C_*, FliC*_S.T_*, and FliC*_P.A_*. The N-terminal and C-terminal regions are identified in green, while the central regions are hypervariable and shown in blue or transparent color. The TLR5 activation hot spots (89–96th amino acids) are highlighted with a red rectangle. Another site related to TLR5 binding, E114, is marked with a red star. The “-” indicates the omitted amino acid residues. The identities (%) of amino acid sequence (**B**) and TLR5 binding sites sequence (**C**) of FliC*_E.C_*, FliC*_S.T_*, and FliC*_P.A_* were analyzed using DNAStar’s Lasergene software MegAlign program, respectively. (**D**–**F**) The secondary structure of the FliC monomer was modeled by PyMol in both cartoon and surface patterns. The D0 and D1 domains are indicated in green and blue, respectively. The D2 and D3 domains are painted yellow. The TLR5 binding sites (89–96th) are labeled in red. The residues 89, 90, 94, and 114 are marked in orange, cyan, gray, and purple, respectively.

**Figure 2 vaccines-12-01212-f002:**
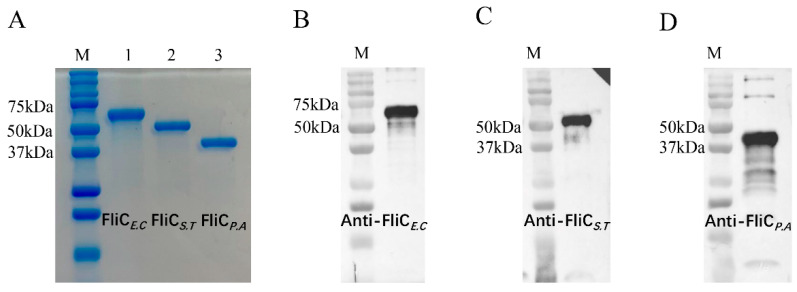
Purification and identification of FliC*_E.C_*, FliC*_S.T_*, and FliC*_P.A_*. (**A**) The purified FliC*_E.C_*, FliC*_S.T_*, and FliC*_P.A_* were stained with Coomassie Blue to present the SDS-PAGE profile. (**B**–**D**) The protein bands were individually detected with anti-FliC*_E.C_*, anti-FliC*_S.T_*, and anti-FliC*_P.A_* polyclonal antibodies (1:8000). A second antibody, HRP-conjugated goat anti-mouse IgG (1:10,000), was used for chemiluminescence in Western Blot.

**Figure 3 vaccines-12-01212-f003:**
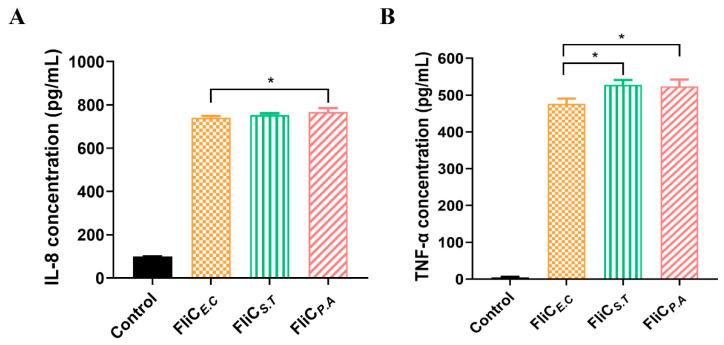
TLR5 activation abilities of FliC*_E.C_*, FliC*_S.T_*, and FliC*_P.A_*. (**A**) The concentration of IL-8 in the Caco-2 cell supernatants. (**B**) The concentration of TNF-α in the Caco-2 cell supernatants. Each sample was repeated three times. The data were expressed as mean ± standard error of the mean (SEM) from the three independent experiments. * *p* < 0.05.

**Figure 4 vaccines-12-01212-f004:**
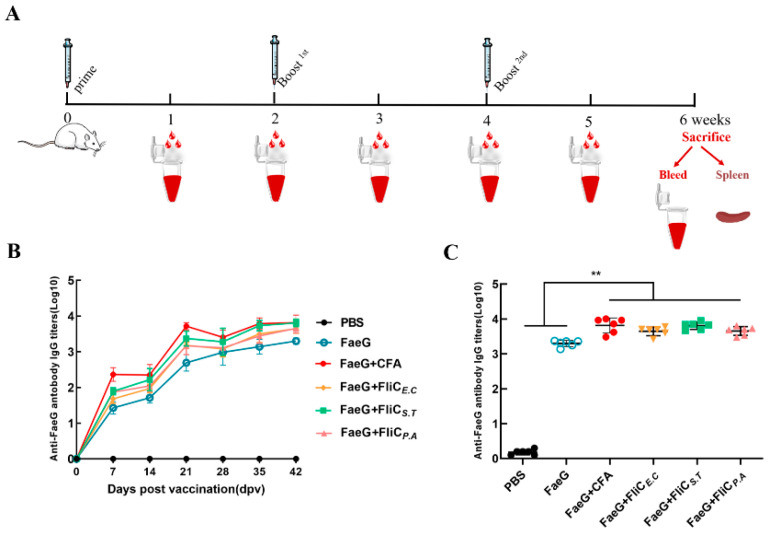
Antibody response induced by FliC*_E.C_*, FliC*_S.T_*, and FliC*_P.A_* recombinant flagellins as adjuvants in mice. (**A**) A schematic of the experimental design for mice immunization. Mice were immunized in weeks 0, 2, and 4. Serum samples were collected every week after the primary immunization. (**B**) The trends and longevity of anti-FaeG IgG production during the immunization course. (**C**) The total anti-FaeG IgG antibody levels in six individual mice after the final immunization. The FaeG + CFA groups were used as a positive adjuvant control. Each icon represents the antibody titer from an individual mouse in the corresponding group. The bars are considered as the mean ± SEM for the immunized or control groups. Significant differences are shown as ** *p* < 0.01.

**Figure 5 vaccines-12-01212-f005:**
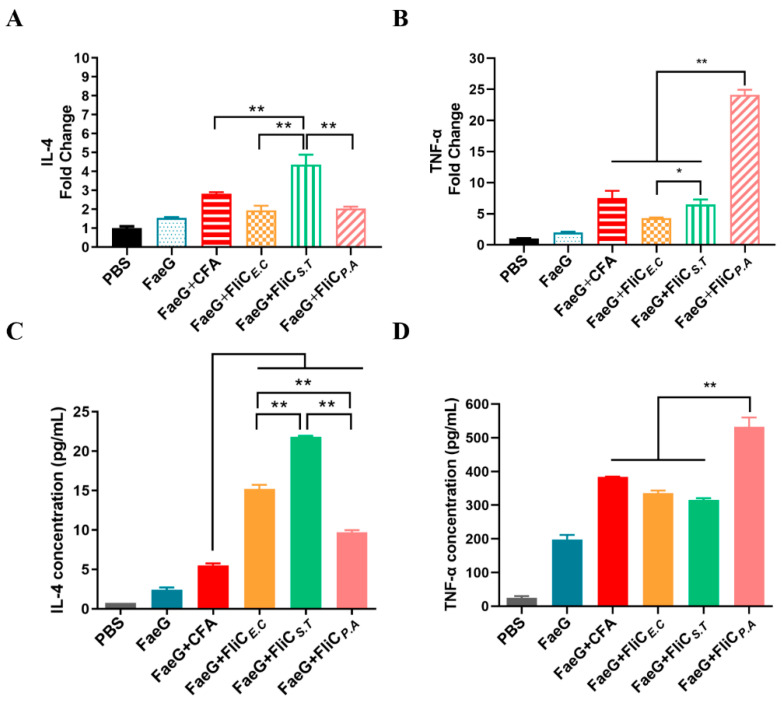
The transcription and translation of IL-4 and TNF-α cytokine production from splenocytes in immunized mice. Splenocytes were stimulated with FaeG antigen protein (5 μg/mL) in vitro for 72 h after isolation. The detection of IL-4 (**A**) and TNF-α (**B**) mRNA transcription levels was performed on re-stimulated splenocytes using qRT-PCR. The concentrations of IL-4 (**C**) and TNF-α (**D**) were determined from the supernatant of splenocytes. The FaeG + CFA groups as positive adjuvant control. Statistical analysis was performed using one-way ANOVA. * *p* < 0.05; ** *p* < 0.01.

**Table 1 vaccines-12-01212-t001:** The primer oligonucleotide sequences used in this study.

Primer Names	Primer Sequences (5′-3′)
*fliC_EC_*-F	CGCGGATCCATGGCACAAGTCATTAATACCAACAG
*fliC_EC_*-R	GAGCGTCGACTTAACCCTGCAGCAGAGACAGAAC
*fliC_ST_*-F	CGCGGATCCATGGCACAAGTCATTAATACA
*fliC_ST_*-R	AACGAGCTCTTAACGCAGTAAAGAGAGGAC
*fliC_PA_*-F	CGCGGATCCATGGCCCTTACAGTCAACACG
*fliC_PA_*-R	AACGAGCTCTTAGCGCAGCAGGCTCAGGA
*faeG-*F	ATTCGGGATCCATGAAAAAGAC
*faeG-*R	CAGCGTCGACTTAGTAATAAGT
*GAPDH*-F	GCCTTCCGTGTTCCTACCC
*GAPDH*-R	TGCCTGCTTCACCACCTTC
*Il4-*F	ACAGGAGAAGGGACGCCAT
*Il4-*R	GAAGCCCTACAGACGAGCTCA
*Tnfα-*F	AGCCCCCAGTCTGTATCCTT
*Tnfα-*R	CTCCCTTTGCAGAACTCAGG

## Data Availability

The data that support the findings of this study are available from the corresponding author upon reasonable request.

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
