# Peer review of "Differential Adjuvant Activity by Flagellins from Escherichia coli, Salmonella enterica Serotype Typhimurium, and Pseudomonas aeruginosa"

_vaccines, 2024, doi:10.3390/vaccines12111212_

Round 1
Reviewer 1 Report
Comments and Suggestions for Authors
This manuscript describes the comparison of the adjuvant activity of recombinant flagellins from Escherichia coli, Salmonella Typhimurium and Pseudomonas Aeruginosa. The paper is well written and easy to follow. It is within the scope of Vaccines, as it deals with a topical issue - the search for promising adjuvants. However, a number of points need to be corrected and revised.
One particular model antigen, called FaeG, remains largely unknown and uncharacterised to the reader.
As the production and use of recombinant flagellins FliC as vaccine adjuvants is not a new idea, it is recommended that the authors present the current state of the art in this field and highlight the novelty and originality of their research.
The following points should be addressed in the revised manuscript
L124-126. Please, state the source of the anti-FliC mouse sera. How were they obtained prior to isolation of the recombinant proteins?
L 151. The use of the recombinant FaeG protein as a model antigen should be explained in the text. It is combined with antigens being studied for their adjuvant activity and should also be characterized as a target antigen. Information on this protein, its source, methods of production, isolation, purification, identification and testing for pyrogenicity should be provided in the same way as for FliC antigens.
L160. Fig 5A. Mention and discuss results in the Results section. Start numbering the figures discussed in chronological order, starting with FIG 1.
L161. Specify pH of buffers used
L180. 2.5×106 cells is the normal number of splenocytes in the spleen. Does this mean that all the cells from the whole spleen were placed in one well?
L250. Binding of anti-recombinant FliC polyclonal antibody to recombinant FliC is not reliable сonfirmation that the prepared recombinant FliCs are similar to native flagellins. For example, antibody to poly-His tag can bind to all prepared FliCs.
It would be better to confirm it with
1/ Binding to Anti-flagellin or Anti-bacterial flagella antibodies OR.
2/The prepared FliCs should inhibit antibody binding to bacteria OR.
3/ Provide negative controls in immunoblotting (Anti-FlicEC + FliCST/FliCPA etc).
L273-275. Cells were stimulated with 5 μg/mL of purified recombinant FliCE.C, FliCS.T, and FliCP.A
Since FliCP.A is 40kDa vs FliCE.C 60kDa, it has higher molar concentration, therefore higher activity. Significance of differences in Fig 3 should be recalculated. The statement ‘different species exhibit varying TLR5 activation’ in L270 should be revised.
L290. The resume should be revised, since the phrase “the difference was not statistically significant (P>0.05), indicating that FliCS.T has superior adjuvant activity in vivo” is contradictory.
Fig. 4D is redundant because it duplicates Fig. 4C.
Author Response
Manuscript ID: vaccines-3243005 - Major Revisions
Title: "Differential adjuvant activity by flagellins from Escherichia coli, Salmonella enterica serotype Typhimurium, and Pseudomonas aeruginosa"
Dear respected reviewer,
Warmly greetings from the authors!
We would like to thank you very much for your nice comments and suggestions on our manuscript, you raised helped us improve our revised manuscript. As the your comments and requests, accordingly, we have revised the manuscript wherever necessary and highlighted them in the revised manuscript with yellow. Further to your comments and suggestions, the corrections in the manuscript have been made and the responses to these comments are included below. Meanwhile, please find the enclosed copy of the revised manuscript.
- One particular model antigen, called FaeG, remains largely unknown and uncharacterised to the reader.
Response: The FaeG protein, used as a model antigen in this study, is the major subunit of F4 fimbriae, which are essential for the adhesion of porcine enterotoxigenic Escherichia coli (ETEC) that produce F4 fimbriae to host intestinal epithelial cells. The characterization of the FaeG antigen has been added in the revised manuscript (Line 166-169).
- As the production and use of recombinant flagellins FliC as vaccine adjuvants is not a new idea, it is recommended that the authors present the current state of the art in this field and highlight the novelty and originality of their research.
Response: The broad development involves the use of flagellin fusion proteins or their combination with antigens to exert adjuvant effects. This approach has been widely utilized in developing vaccines for influenza, tuberculosis, and cancer, among others. Furthermore, flagellin-based vaccines have shown promise in generating mucosal immunity by stimulating IgA production in mucosal tissues via mucosal routes. However, clinically approved vaccines that utilize flagellin as an adjuvant have not yet been authorized. These adjuvants are currently being tested in preclinical and clinical trials, such as VAX125 (STF2-HA in an influenza vaccine candidate) and STF2.4xM2e (M2e fused to the C-termi-79 nus of the full-length sequence of Salmonella enterica serotype Typhimurium flagellin 80 phase 2 (fljB) in influenza vaccines). Nevertheless, both of these flagellin adjuvants in clinical trials were derived from S. Typhimurium, and the specific effects of flagellin adjuvants from other species of bacteria remain unclear. The above content regarding the current state of the art in using flagellin as adjuvants has been added to the introduction section of the revised manuscript (Line 72-81).
- L124-126. Please, state the source of the anti-FliC mouse sera. How were they obtained prior to isolation of the recombinant proteins?
Response: The anti-FliC mouse sera were obtained by immunizing mice with crude extracts of flagellin derived from E. coli O157 (EHEC) EDL933, S. Typhimurium, and P. aeruginosa, following the protocol described by Duan et al. (2013). Briefly, bacteria were cultured overnight in LB medium at 37°C with aeration (80 rpm). The cells were washed with phosphate-buffered saline, and the flagella were detached by subjecting the cultures to four rounds of shearing. The bacteria were centrifuged at 4,900 ×g for 30 minutes to remove outer membranes and bacterial debris. The cell-free supernatants were then ultracentrifuged at 22,800 ×g for 1 hour at 4°C. After acetone precipitation, the flagellar filaments were collected from the supernatant and resuspended in PBS. Endotoxins were removed from the crude flagellin using Pierce™ High-Capacity Endotoxin Removal Resin, and the purified flagellin was used as the immunizing antigen, combined with Freund’s adjuvant. The mice sera were collected after three vaccinations administered at bi-weekly intervals. The sera were purified using protein A affinity. The above information has been incorporated into the revised manuscript (Line 136-139).
- L 151. The use of the recombinant FaeG protein as a model antigen should be explained in the text. It is combined with antigens being studied for their adjuvant activity and should also be characterized as a target antigen. Information on this protein, its source, methods of production, isolation, purification, identification and testing for pyrogenicity should be provided in the same way as for FliC antigens.
Response: The FaeG protein, utilized as a model antigen in this study, is derived from the major subunit of F4 fimbriae, which are essential for the adhesion of ETEC to host intestinal epithelial cells. FaeG was selected due to its status as a well-characterized antigen, known to elicit immune responses, making it an appropriate candidate for evaluating the adjuvant efficacy of flagellin proteins. The FaeG protein was expressed in E. coli using a pET28a expression system and purified via nickel-affinity chromatography. Full details regarding FaeG production and characterization, including the purification method and confirmation through SDS-PAGE, are consistent with those provided for the flagellins. The characterization of the FaeG antigen has been incorporated into both the Materials and Methods (Line 108,112) and Discussion (Line 387-389) sections of the revised manuscript. The SDS-PAGE and Western Blot (WB) results of the purified FaeG recombinant protein are shown below.
- Fig 5A. Mention and discuss results in the Results section. Start numbering the figures discussed in chronological order, starting with FIG 1.
Response: We appreciate your constructive suggestions. Fig. 5A was mentioned in the Results section of the original manuscript. In order to enhance the clarity and flow of the manuscript for readers, we have implemented your recommendation to renumber all figures according to the chronological order in which they are first mentioned in the text, starting with Fig. 1.
- Specify pH of buffers used
Response: The pH values of PBST and the carbonate buffer have been included in the revised manuscript (Line 180,183).
- 2.5×106 cells is the normal number of splenocytes in the spleen. Does this mean that all the cells from the whole spleen were placed in one well?
Response: Thank you for your insightful question. When we mention 2.5 × 106 splenocytes as a working number, this refers to the number of cells placed in a specific well of a 6-well cell plate, rather than the total number of splenocytes from the entire spleen. In our experiment, the total splenocyte count from a mouse spleen typically ranges between 2.0 × 107 and 2.5 × 107. Moreover, to minimize individual differences between groups, we pooled the splenocytes isolated from 6 mice per group and then performed cell counting. Afterward, we plated the cells in a 6-well cell culture plate, with a cell density of 2.5 × 106 per well. Therefore, the 2.5 × 106 cells represent only a portion of the total splenocyte population, isolated for use in the experiment.
I hope this clarifies the situation, and I would be happy to provide further details if needed.
- Binding of anti-recombinant FliC polyclonal antibody to recombinant FliC is not reliable сonfirmation that the prepared recombinant FliCs are similar to native flagellins. For example, antibody to poly-His tag can bind to all prepared FliCs.
It would be better to confirm it with
1/ Binding to Anti-flagellin or Anti-bacterial flagella antibodies OR.
2/The prepared FliCs should inhibit antibody binding to bacteria OR.
3/ Provide negative controls in immunoblotting (Anti-FlicEC + FliCST/FliCPA etc).
Response: Thank you for your constructive suggestions. In fact, the anti-FliC mouse sera were obtained by immunizing mice with crude extracts of flagellin derived from EHEC EDL933, S. Typhimurium, and P. aeruginosa strains, not from mice immunized with recombinant FliC proteins. The source and detailed information regarding these antibodies have been addressed in response to question 3. Therefore, we believe the WB result serves as a reliable confirmation.
- L273-275. Cells were stimulated with 5 μg/mL of purified recombinant FliCC, FliCS.T, and FliCP.A. Since FliCP.A is 40kDa vs FliCE.C 60kDa, it has higher molar concentration, therefore higher activity. Significance of differences in Fig 3 should be recalculated. The statement ‘different species exhibit varying TLR5 activation’ in L270 should be revised.
Response: Thank you for your insightful comments regarding the molar concentrations of FliCE.C, FliCS.T, and FliCP.A. We acknowledge that the molecular weights of these flagellins differ, with FliCP.A (40 kDa) being smaller than FliCE.C (60 kDa). However, the primary objective of our study was to evaluate the immunoadjuvant activities of these flagellins at an equal mass concentration of 5 μg/mL, which is a common practice in comparative studies of protein adjuvants. This approach allows for a direct assessment of their immunological effects under consistent dosing conditions, relevant to potential applications in vaccine formulations.
While it is true that molar concentrations can influence biological activity, we believe that using equal mass concentrations provides a more practical comparison of their effects in vivo. Recalculating significance based on molar concentration may not yield additional insights pertinent to our study objectives, as clinical applications typically rely on dosage by weight rather than molarity.
Regarding the statement on line 270, we maintain that “different species exhibit varying TLR5 activation” is appropriate, as our results clearly demonstrate distinct biological responses to these flagellins at equal mass concentrations.
Thank you once again for your valuable feedback.
L290. The resume should be revised, since the phrase “the difference was not statistically significant (P>0.05), indicating that FliCS.T has superior adjuvant activity in vivo” is contradictory.
Response: Thank you for your thoughtful suggestion. The sentence has been rephrased in the revised manuscript as follows: “Although the FliCS.T adjuvant group displayed a higher anti-IgG antibody titer (3.81±0.11) than the FliCE.C (3.65±0.11) and FliCP.A (3.66±0.11) groups, the difference was not statistically significant (P>0.05) (Fig. 4C). These findings indicate that FliCS.T has superior adjuvant activity in vivo.” in the revised manuscript.
Fig. 4D is redundant because it duplicates Fig. 4C.
Response: Thank you for your valuable suggestion. Fig. 4D has been removed from the revised manuscript.
Best Regards,
Yours sincerely,
Qiangde Duan, Ph.D,
Professor of Veterinary Microbiology
Yangzhou University College of Veterinary Medicine
Yangzhou, 225009, China
Tel: (0086)-514-87972590
Fax: (0086)-514-87311374
E-mail address: dqd@yzu.edu.cn
Reviewer 2 Report
Comments and Suggestions for Authors
The topic in this manuscript is interesting, and the investigators started with bioinformatics analysis and then tested the hypothesis in the animal model. The authors are encouraged to address the questions below.
1) To judge the efficacy of an experimental adjuvant, it is ideal for the authors to include positive control- a type of successful adjuvant that is well-recognized in the field. Then, the authors can compare the performance among these three flagellins, and with the well-recognized adjuvant. This is obvious pitfall of this study in this reviewer's opinion.
2) The authors measured the antibody responses and the expression of IL-4 and TNF-alpha. Then, it is concluded that FliCS.T and FliCP.A are more effective in inducing T-cell responses. Please indicate what are the standard well to indicate the level of antibody response and the cytokine/T cell response regarding the performance of adjuvants. Again, if a positive control of a successful adjuvant is used in this work, these levels can be compared.
3) The panel A of the figure 1 is the key for this work. We can see a huge level of differences in the amino acid sequences among these three bacterial species. Besides, there is likely a huge level of polymorphisms among each of the bacterial strains. For example, more than 95% of E. coli strains are not pathogenic, and there are many other strains and groups of pathogenic E. coli. This reviewer is not convinced of the conclusion made from the alignment of AA sequences in Panel-A of Figure 1. Please be sure the foundation of this work based on this sequence analysis is solid.
Comments on the Quality of English LanguagePlease pay attention to the writing of the names of the bacteria.
For example, Typhimurium is a serovar for Salmonella, and should be written in a regular text, and "T" needs to be capitalized. In the manuscript title, Aeruginosa should be as aeruginosa.
IL-4 should be used to replace li4 in the manuscript.
Please carefully check the entire manuscript, and there are lots of typos.
The authors are encouraged to polish the manuscript. For example, in Abstract:
line 15: what does "their" here refer to?
Lines 19 and 25: when you write the mouse model, in vivo can be removed, which is redundant.
Line 32: remove "adjuvant" since you have "adjuvant activity"
Author Response
Manuscript ID: vaccines-3243005 - Major Revisions
Title: "Differential adjuvant activity by flagellins from Escherichia coli, Salmonella enterica serotype Typhimurium, and Pseudomonas aeruginosa"
Dear respected reviewer,
Warmly greetings from the authors!
We would like to thank you very much for your nice comments and suggestions on our manuscript, you raised helped us improve our revised manuscript. As the your comments and requests, accordingly, we have revised the manuscript wherever necessary and highlighted them in the revised manuscript with yellow. Further to your comments and suggestions, the corrections in the manuscript have been made and the responses to these comments are included below. Meanwhile, please find the enclosed copy of the revised manuscript.
- To judge the efficacy of an experimental adjuvant, it is ideal for the authors to include positive control- a type of successful adjuvant that is well-recognized in the field. Then, the authors can compare the performance among these three flagellins, and with the well-recognized adjuvant. This is obvious pitfall of this study in this reviewer's opinion.
Response: Thank you very much for your valuable suggestion. As Freund’s adjuvant is a well-recognized adjuvant in the field, we had indeed included a Freund’s adjuvant group as a positive control during the experimental design. However, since the primary goal of this study was to provide a direct comparison of the adjuvant activities of the three flagellins from different bacterial species, we initially chose not to present the corresponding data and results. In light of your suggestion, we have now included the data in the revised manuscript to allow for a more comprehensive comparison. The updated results have been incorporated into the results section, and the newly drafted Figures 4 and 5 have been updated accordingly.
- The authors measured the antibody responses and the expression of IL-4 and TNF-alpha. Then, it is concluded that FliCT and FliCP.A are more effective in inducing T-cell responses. Please indicate what are the standard well to indicate the level of antibody response and the cytokine/T cell response regarding the performance of adjuvants. Again, if a positive control of a successful adjuvant is used in this work, these levels can be compared.
Response: Thank you for your valuable comment and suggestion. In this study, the antibody levels against the model antigen FaeG were used as a marker of the ability of the three flagellins to induce humoral immunity, while the expression levels of IL-4 and TNF-α were measured to assess their ability to induce cellular immunity, specifically T-cell responses. Based on your suggestion, we have incorporated Freund’s adjuvant, emulsified with the model antigen, as a positive control to benchmark the immunoadjuvant effects, and PBS and FaeG alone as negative controls. This adjustment provides a more reliable comparison of the immunoadjuvant performance of the three flagellins in vivo. The updated results have been included in newly drafted Figures 4 and 5 in the revised manuscript.
- The panel A of the figure 1 is the key for this work. We can see a huge level of differences in the amino acid sequences among these three bacterial species. Besides, there is likely a huge level of polymorphisms among each of the bacterial strains. For example, more than 95% of coli strains are not pathogenic, and there are many other strains and groups of pathogenic E. coli. This reviewer is not convinced of the conclusion made from the alignment of AA sequences in Panel-A of Figure 1. Please be sure the foundation of this work based on this sequence analysis is solid.
Response: We appreciate your insightful comment regarding potential polymorphisms in bacterial strains. In our study, we compared and analyzed the amino acid sequences of flagellins derived from E. coli, S. Typhimurium, and P. aeruginosa. We observed significant differences, particularly in the D2 and D3 domains, while the highly conserved D0 and D1 domains, which are critical for TLR5 activation, also exhibited some variations. This suggests that flagellins from different bacterial species may exhibit varying adjuvant activities.
- coli is part of the normal microbiota in the intestines of humans and animals, with only a small subset of serotypes being pathogenic. There are currently 53 known H serotypes of E. coli. One of our previous studies demonstrated that flagellin proteins derived from three different E. coli serotypes (H1, H7, H19) exhibited the same immunoadjuvant activity both in vitro and in vivo, suggesting that flagellins from different serotypes within the same bacterial species may possess similar immunoadjuvant properties (Pang et al., 2021, DOI: 10.1186/s12917-022-03412-3). Given that E. coli has up to 53 H serotypes, future work will focus on comparing the immunoadjuvant activities of flagellins from additional, and potentially all, serotypes to further substantiate this conclusion.
- Comments on the Quality of English Language. Please pay attention to the writing of the names of the bacteria. For example, Typhimurium is a serovar for Salmonella, and should be written in a regular text, and "T" needs to be capitalized. In the manuscript title, Aeruginosa should be as aeruginosa. IL-4 should be used to replace il4 in the manuscript.
Response: Thank you for your insightful comments regarding the quality of the English language and the correct formatting of scientific terminology.
Bacterial Names: We have corrected the formatting of the bacterial names throughout the manuscript. Specifically, "Typhimurium" is now written in regular text as a serovar for Salmonella, with "T" capitalized, and Salmonella enterica serotype Typhimurium is now formatted correctly. In the manuscript title, "Aeruginosa" has been changed to aeruginosa (lowercase "a") to follow the proper nomenclature for Pseudomonas aeruginosa.
Cytokine Abbreviations: The incorrect use of "il4" has been replaced with IL-4 throughout the manuscript to ensure consistency with standard scientific conventions.
We have reviewed the entire manuscript to ensure the accurate use of scientific names and abbreviations, and made all necessary corrections. Thank you for your attention to these details, which has greatly improved the precision and clarity of our manuscript.
- Please carefully check the entire manuscript, and there are lots of typos.
The authors are encouraged to polish the manuscript. For example, in Abstract:
Line 15: what does "their" here refer to? Lines 19 and 25: when you write the mouse model, in vivo can be removed, which is redundant. Line 32: remove "adjuvant" since you have "adjuvant activity"
Response: Thank you for your thorough review and constructive feedback. We sincerely appreciate your detailed comments and suggestions. We have carefully revised the manuscript to address the issues you raised:
Line 15: The pronoun "their" refers to flagellins from various bacterial species, specifically denoting their immunoadjuvant effects. We have clarified this in the revised version.
Lines 19 and 25: You are correct that "in vivo" is redundant when referring to the mouse model. We have removed "in vivo" from these lines to improve clarity and avoid redundancy.
Line 32: We agree with your suggestion to remove "adjuvant" in "adjuvant activity," as it is redundant. The term "activity" now stands alone for clarity.
Additionally, we have performed a thorough proof reading of the entire manuscript to correct any remaining typos and polished the text to enhance readability and academic rigor. Thank you again for your valuable input, which has significantly improved the quality of our manuscript.
Best Regards,
Yours sincerely,
Qiangde Duan, Ph.D,
Professor of Veterinary Microbiology
Yangzhou University College of Veterinary Medicine
Yangzhou, 225009, China
Tel: (0086)-514-87972590
Fax: (0086)-514-87311374
E-mail address: dqd@yzu.edu.cn
Round 2
Reviewer 1 Report
Comments and Suggestions for Authors
The authors provided the necessary explanations and comments to the reviewer's questions and made the necessary corrections, which improved the manuscript.
Author Response
Reviewer 1#
- The authors provided the necessary explanations and comments to the reviewer's questions and made the necessary corrections, which improved the manuscript.
Response: Thank you. There are no more comments that need to be answered.
Reviewer 2 Report
Comments and Suggestions for Authors
This reviewer thank the authors for their hard work in addressing my comments. I can tell that the quality of this manuscript has been improved after the revision. However, it seems the authors failed to address two key questions. I am here to reiterate these two questions.
Question-1: the title and conclusion of this work are related to three organisms. As figure 1A tells, there is a great level of amino acid diversity among the sequences of these three bacterial species. How conserved is the amino acid sequence within each of these three bacteria species?
Question-2: the authors responded that the positive control, Freunds' adjuvant, was added in this version of this manuscript. Regarding the positive control, I could not find any changes in the figures, figure legends, or text for the results. please remind me of the changes in this revised manuscript.
Lines 102-103: please use italicized text for bacteria. Thanks!
Author Response
Manuscript ID: vaccines-3243005 - Major Revisions
Title: "Differential adjuvant activity by flagellins from Escherichia coli, Salmonella enterica serotype Typhimurium, and Pseudomonas aeruginosa"
Dear respected reviewer,
Thank you very much for your valuable comments and suggestions on our manuscript. Your feedback has greatly helped us improve the quality of the revised version. In response to your comments and requests, we have carefully revised the manuscript where necessary and re-highlighted the changes in green. We have addressed each of your comments in detail, and the corresponding responses are provided below. Please find the revised manuscript enclosed for your consideration.
Reviewer 2#
- The title and conclusion of this work are related to three organisms. As Figure 1A tells, there is a great level of amino acid diversity among the sequences of these three bacterial species. How conserved is the amino acid sequence within each of these three bacteria species?
Response: Thank you very much for your insightful comments. In response to your query, we conducted a sequence alignment and analysis of the core motif (amino acids 89–96) that binds to the TLR5 receptor, focusing on flagellin from 20 different serotypes of E. coli, 7 different serotypes of Salmonella, and 5 different serotypes of P. aeruginosa. The results are shown in the figure below. The NH2- and COOH-terminal regions of flagellin are highly conserved within each bacterial species, with the main sequence differences located in the central hypervariable region. The core motif for TLR5 receptor binding is similarly conserved within each of these three bacteria species.
For E. coli flagellin, the primary variations are at position 91, where valine (V) or isoleucine (I) is found, and at position 95, where serine (S) or threonine (T) appears. In Salmonella flagellin, the only variation is at position 91, with alanine (A) or serine (S) observed. In P. aeruginosa, methionine (M) or isoleucine (I) is found at position 91, while alanine (A) or serine (S) appears at position 95.
Furthermore, in one of our previous studies, we demonstrated that flagellin proteins derived from three different E. coli serotypes (H1, H7, H19) exhibited the same immunoadjuvant activity both in vitro and in vivo, suggesting that flagellins from different serotypes within the same bacterial species may possess similar immunoadjuvant properties (Pang et al., 2021, DOI: 10.1186/s12917-022-03412-3). Whether different flagellin serotypes from Salmonella and P. aeruginosa exhibited the same immunoadjuvant activity needs further study.
(A) The Full-length sequence comparison and homology analysis of TLR5 binding sites in different serotypes of E. coli.
(B) The Full-length sequence comparison and homology analysis of TLR5 binding sites in different serotypes of Salmonella.
(C) The Full-length sequence comparison and homology analysis of TLR5 binding sites in different serotypes of P. aeruginosa.
- 2. The authors responded that the positive control, Freunds' adjuvant, was added in this version of this manuscript. Regarding the positive control, I could not find any changes in the figures, figure legends, or text for the results. please remind me of the changes in this revised manuscript.
Response: Thank you for your valuable feedback. In response to your concerns, we have included data on the positive control group, CFA, in the first revised manuscript (R1) to enhance the comparative analysis. The corresponding updates have been made in the Results section (lines 171-172, lines 296–311, line 318, lines 326–340, and) line 352), which have been highlighted in green for your convenience. Furthermore, the newly revised Figures 4 and 5, along with their corresponding figure legends, have been updated to reflect these changes in the latest version of the manuscript.
lines 171-172:
lines 296–311:
line 318:
lines 326–340:
line 352
- Lines 102-103: please use italicized text for bacteria. Thanks!
Response: Thank you for your suggestion. The bacterial names have now been italicized consistently throughout the revised manuscript.
Lines 102-103:

Round 3
Reviewer 2 Report
Comments and Suggestions for Authors
The authors have properly addressed most of my questions.
Regarding the use of Freund's adjuvants, I can see that the authors did include it in the current study. These changes can be found in M&M, and Results. However, the authors MUST summarize the related outcomes in the results section and provide a meaningful conclusion.
Author Response
Manuscript ID: vaccines-3243005 - Minor Revisions
Title: "Differential adjuvant activity by flagellins from Escherichia coli, Salmonella enterica serotype Typhimurium, and Pseudomonas aeruginosa"
Dear respected reviewer,
Thank you again for your valuable suggestions on our manuscript. In response to your suggestions, we have carefully revised the result section and re-highlighted the changes in yellow. We have addressed your comment in detail, and the corresponding responses are provided below. Please find the revised manuscript enclosed for your consideration.
1.Regarding the use of Freund's adjuvants, I can see that the authors did include it in the current study. These changes can be found in M&M, and Results. However, the authors MUST summarize the related outcomes in the results section and provide a meaningful conclusion.
Response: Thank you very much for your insightful comments. In response to your request, we have summarized the related outcomes in Sections 3.4 and 3.5 of the Results. Additionally, we have included a meaningful conclusion regarding these findings in the revised Conclusion section of the manuscript. We believe these revisions enhance the clarity and impact of our study.
Best Regards,
Yours sincerely,
Qiangde Duan, Ph.D,
Professor of Veterinary Microbiology
Yangzhou University College of Veterinary Medicine
Yangzhou, 225009, China
Tel: (0086)-514-87972590
Fax: (0086)-514-87311374
E-mail address: dqd@yzu.edu.cn